# Polymeric Nanoparticles for Inhaled Vaccines

**DOI:** 10.3390/polym14204450

**Published:** 2022-10-21

**Authors:** Nusaiba K. Al-Nemrawi, Ruba S. Darweesh, Lubna A. Al-shriem, Farah S. Al-Qawasmi, Sereen O. Emran, Areej S. Khafajah, Muna A. Abu-Dalo

**Affiliations:** 1Department of Pharmaceutical Technology, Faculty of Pharmacy, Jordan University of Science and Technology, P.O. Box 3030, Irbid 22110, Jordan; 2Department of Chemistry, Faculty of Science and Art, Jordan University of Science and Technology, P.O. Box 3030, Irbid 22110, Jordan

**Keywords:** inhaled vaccine, vaccine, immunization, nanoparticles, polymeric

## Abstract

Many recent studies focus on the pulmonary delivery of vaccines as it is needle-free, safe, and effective. Inhaled vaccines enhance systemic and mucosal immunization but still faces many limitations that can be resolved using polymeric nanoparticles (PNPs). This review focuses on the use of properties of PNPs, specifically chitosan and PLGA to be used in the delivery of vaccines by inhalation. It also aims to highlight that PNPs have adjuvant properties by themselves that induce cellular and humeral immunogenicity. Further, different factors influence the behavior of PNP in vivo such as size, morphology, and charge are discussed. Finally, some of the primary challenges facing PNPs are reviewed including formulation instability, reproducibility, device-related factors, patient-related factors, and industrial-level scale-up. Herein, the most important variables of PNPs that shall be defined in any PNPs to be used for pulmonary delivery are defined. Further, this study focuses on the most popular polymers used for this purpose.

## 1. Introduction

Pulmonary drug delivery is an attractive route for drug administration and targeting, especially in comparison with intravenous injection. Pulmonary drug delivery is a non-invasive technique that uses accessible large mucosal surface areas for rapid absorption and activation [1]. Pulmonary delivery in the form of dry powder inhalations (DPI) maintains vaccine constancy and integrity. High density of antigen-presenting cells (APC) such as alveolar macrophages (AMs) and dendritic cells (DCs) in the lung serves as an ideal target for the antigen to stimulate a strong immune response that results in mucosal and systemic immunity [2]. Therefore, pulmonary drug delivery has gained substantial interest over the past few decades.

Most pathogens enter the body via mucosal surfaces. As such, mucosal vaccination can be used to considerably improve the mucosal immune response. Vaccine administration through mucosal surfaces such as oral, rectal, vaginal, nasal, intranasal or pulmonary tissue can effectively trigger mucosal immune response [3]. As such, mucosal vaccines are feasible for large-scale vaccination and eliminate the risk of blood-borne infections posed by injected vaccines [4]. The benefits of mucosal vaccines also include convenient distribution and administration as well as improved patient compliance [5,6].

Unfortunately, mucosal vaccines still face challenges that include low delivery of preventive viral epitopes and inadequate humoral and cell-mediated response. Mucosal vaccines are also inefficient adjuvants when used in certain protocols [7]. The typical delivery carrier for vaccines should have high efficiency, low cytotoxicity without harmful effects to normal cells, good reproducibility and easy preparation [4,8]. Polymeric nanoparticles (PNPs) have been identified as an ideal carrier system for vaccine delivery that can fulfill these important requirements [9].

Recently, biocompatible and biodegradable PNP use has increased for pharmaceutical delivery including vaccines [10]. PNPs can incorporate different antigens and deliver them to APCs. Antigens are protected from degradation by different enzymes and provide controlled antigen release, which may reduce the number of required doses [8,10]. In addition, PNPs have demonstrated adjuvant properties such as inducing cellular and humeral antigen immunogenicity [11,12]. Other advantages include the fact that PNPs are non-viral vectors, non-immunogenic, biocompatible and have a large specific surface area [8]. PNP absorption by APCs have also been shown to not only induce but also increase an effective immune response [8,11]. Finally, many polymers such as chitosan and poly (D, L-lactide-co-glycolide) (PLGA) are compatible with vaccine loading [13,14].

In this review, the latest research on vaccine delivery by the pulmonary route us-ing PNPs is summarized. This includes an overview of the most used polymers in in-haled vaccinations along with major PNP applications. Further, the most important parameters that are required to define these systems are reviewed. Finally, the challenges and limitations of formulating inhaled vaccines loaded in PNPs will be discussed. In summary, this is the first comprehensive review in the field of using PNPs to deliver vaccines pulmonary. It would be a good starting point for any researcher seek-ing work in this field.

## 2. Nanoparticles for Inhaled Vaccines

### 2.1. Polymeric Nanoparticles and Drug Delivery

Recently, PNPs have become the focus of medical application development because of their simplicity of preparation and design, biocompatibility, and variety of structures. Further, PNPs show enhanced efficacy and bioavailability compared with conventional drugs. Their ability to transport active ingredients to the targeted tissue or organ without affecting the drug stability and in higher concentrations made them favorable over other formulations. Moreover, PNPs can be used to control, delay, or sustain drug release.

PNPs are particles with a size range of 10–100 nm. Although 100 nm-size nanoparticles (NPs) offer the advantage of high-efficient intracellular uptake, NPs larger than 100 nm are preferred for their higher drug loading ability [15]. NPs have a high surface area-to-volume ratio, making them appropriate for drug delivery applications [1,16,17].

PNPs can be classified into two main types: nanocapsules and nanospheres. Nanocapsules act as a reservoir for drug retention in an aqueous or oily liquid in the vesicle core enclosed by a solid polymeric shell. Meanwhile, nanospheres are defined as a solid matrix polymer in which molecules are either trapped in the sphere center or adsorbed at the nanoparticle surface [18,19,20].

Polymers are the main component used in PNP formulation. Both natural and synthetic polymers have been used in PNP formulation that allow for degradation or metabolization over time in biological systems. The polymer properties affect the overall physicochemical properties and behavior of the PNP carriers [21,22]. The choice in polymer is critical to ensure the safety, efficacy, biodegradability, toxicity, encapsulation efficiency, stability, cost and availability of the drug delivery system [23]. For example, natural polymers (e.g., cyclodextrin) used in preparing PNPs release the imbibed drug faster than synthetic polymers (e.g., PLGA) that provide sustained release over several weeks [24,25,26]. The most commonly used natural polymers for PNP formulation are sodium alginate, gelatin, albumin and chitosan. On the other hand, polylactides (PLAs), polyglycolides (PGAs), PLGAs, polycaprolactone (PCLs), polyanhydrides, polycyanoacrylates, poly (malic acid) (PMLA), polyorthoesters (POEs), polyglutamic acid (PGA), poly (vinyl alcohol) (PVA), poly (N-vinyl pyrrolidone) (PVP), polyacrylamide (PAM), poly (methyl methacrylate) (PMMA), polyacrylic acid (PAA or Carbomer), polyethylene glycol (PEG) and poly (methacrylic acid) (PMAA) are the major synthetic polymers used for PNP formulation [26,27].

PNPs can be prepared using a variety of methods, including solvent evaporation, supercritical fluid, nanoprecipitation technology, salting-out, dialysis techniques and multiple emulsions [28]. The method of preparation and controlling the experimental conditions also influence the formed PNP properties along with their body performance. PNPs are currently used to treat, prevent and diagnose diseases [29,30]. These PNPs have been used to load different pharmaceuticals and target other tissues in the body. Specifically, PNPs have been used for cancer therapy, vaccine delivery and targeted antibiotic delivery [31,32]. The details of these medical NP applications have been discussed in detail in many reviews [33,34].

Unluckily, PNPs are facing many challenges in various aspects, such as the use of high amounts of emulsifier. Emulsifier-free or surfactant-free emulsion is now a hot topic in the PNP industry where green procedures that do not rely on chemical emulsifiers are used [35]. These procedures often use reagents consisting of monomers (mostly acryl or vinyl monomers) and a water-soluble initiator (ionizable initiator) to stabilize the formed PNPs. Other researchers have applied the principles of nucleation and particle growth mechanisms without using emulsifiers [27,36]. Additionally, the use of natural emulsifiers derived from plants, bacteria and fungi have also been used to eliminate synthetic harmful emulsifiers [37].

Unfortunately, many factors essential to these procedures are still uncontrolled and require attention. Moreover, the scale-up process for industrial production of these green products is another problem. Both clinical and pharmaceutical outcomes of lab formulations are subject to alteration during the scale-up process [38]. Reproducibility is another challenge that faces green synthesis of manufacturing PNPs. Some technologies, including supercritical fluid technology, microfluidizer and membrane extrusion technology, have promising scale-up competencies, but only a small number of products produced by these technologies have reached the market [39].

The regulatory requirements for the potential PNPs, including those prepared by green synthesis, are also considered a challenge. The FDA, EMA and other regulatory agencies around the world inspect new PNPs on product-by-product basis. For Investigational New Drug (IND) applications, the preclinical and clinical validation review are mandated by FDA. The appropriate identification includes structure, quality, purity, synthesis methodology, etc. To ensure the efficacy and safety of nanoparticles, additional data such as nanoparticle morphology, size, size distribution, shape, surface additives, specific physic-chemical information and coating effect should be also detailed. PNPs that have successfully reached the market use PNPs, PLGA NPs (e.g., Neulasta^®^ and Copaxone^®^, Macugen^®^ (Bausch & Lomb, Laval, QC, Canada), Eligard^®^ (Tolmar, CO, USA), PegIntron^®^ (Merck, NJ, USA) and Pegasys^®^ (Genentech, CA, USA).

### 2.2. Nanoparticles Drug Delivery to the Lungs

The delivery of particles to the different regions of the lungs depends on the particle size of the formulation. Based on the particle size, there are three different mechanisms of drug deposition through the pulmonary route, impaction, sedimentation, and diffusion [40].

In impaction, the aerosol particles go through the oropharynx and upper respiratory passages at a very high velocity. The particles then interact with the respiratory wall and are deposited in the oropharynx regions [41]. This mechanism can be observed with particle sizes greater than 5 µm mainly in dry powder inhalation (DPI) and metered dose inhalators (MDI) [42]. In the DPI, the deposition is mainly affected by the inspiratory effort of the patient. If the force of inhalation is insufficient, the dry powder will be deposited in the upper airways due to the mass of the particles [42]. In the MDI, high particle sizes also tend to lead to the deposition of the particles in the upper respiratory tract region despite the high speed of the generated aerosol [42,43].

Gravitational forces are mainly responsible for the second mechanism, which is particle sedimentation. Particles with certain mass and sizes between 1 to 5 µm are deposited in the smaller airways and bronchioles [44]. Sedimentation is also influenced by the breathing mechanism; slow breathing patterns provide a sufficient period for efficient sedimentation [45].

The diffusion process plays a major role in the deeper alveolar areas of the lungs. The Brownian motion of the surrounding molecules in the aqueous lung surfactant causes a random movement of the particles that leads to the dissolution of the drug in alveolar fluid when in contact with the lung surfactant which is essential for diffusion [46]. In addition, the diffusion process is also affected by concentration gradient [47]. Particles smaller than one to 0.5 µm are mainly deposited in the alveolar region, while most of the particles, because of their smaller sizes, are exhaled [47].

Moreover, depending on the location of deposition which is mainly affected by the particle size [43], the nanoparticles can interact with different cell types within the respiratory tract, such as epithelial cells and antigen-presenting cells.

The epithelial cells are tightly connected by intercellular junctions called tight junctions. Nanoparticles can pass the respiratory epithelia by two different pathways: through tight junctions between the cells or transcellularly by endocytosis [48].

The firmly sealed tight junctions in the epithelial cells make a barrier for particle permeation [49]. Moreover, the mucus layer, which covers the upper and central respiratory tract, as well as the clearance process in these regions create more barriers that reduce the uptake of nanoparticles in the respiratory lumen [49]. Therefore, agents, called penetration enhancers, which reversibly open the tight junctions are added to the formulations to enhance the transport of particles to the systemic and/or lymphatic circulation [50]. On the contrary, in the distal airway’s epithelium, just before the alveoli, the tight junctions between the epithelial cells are loose and particles up to 22 kDa can passively diffuse via paracellular pathways [50].

In addition to the paracellular route by which relatively small proteins are absorbed, larger proteins can be taken up from the respiratory tract by the transcellular pathway, which includes both nonspecific and specific (receptor-mediated) endocytosis [51]. The transport of antibodies and plasma proteins such as albumin across the epithelial cells occurs by receptor-mediated endocytosis [52]. On the other hand, it has been shown that macromolecular therapeutics pass the epithelial cells by nonspecific endocytosis [51].

There are a variety of immune cell populations in the lungs such as phagocytic cells (macrophages) and antigen-presenting cells (dendritic cells DCs) [53]. The main role of immature DCs, primarily within the mucosal tissue, is to recognize antigens by their protrusions into the airway or alveolar lumen [54]. In this mechanism, depending on the nature of the antigen, DCs get activated by their recognition receptors and enter the maturation process; once maturated, DCs rapidly migrate to the lymph nodes [55]. In the case of maturation, antigens are processed by the DCs and presented to naive T cells by the major histocompatibility complex (MHC II) in combination with upregulated co-stimulatory molecules on the DCs surface (CD40, CD80, CD86) and the release of cytokines [55,56]. The type and combination of the cytokines released will determine the nature of effector T cells induced (Th1, Th2, Treg, and Th17) [55,57].

NPs-based vaccination protocols that mainly target DCs are efficient and promising strategies for the induction and enhancement of immune responses for cancer treatment [58,59] in addition to viral and microbial infections [60,61]. NPs can extend the antigen exposure to immune cells, facilitate antigen capture by DCs and initiate antigen presentation pathways within these cells, thus enhancing T cell-mediated immune responses [62]. In addition, NPs delivery system could be used to enhance DCs activation by triggering cell-surface molecules. This mechanism enhances the internalization of NPs by DC and thus induces immune activation with separate and specific DCs-activating signaling pathways [63].

#### 2.2.1. Nanoparticles through Different Pulmonary Routes of Administration

Nanoparticles are of great interest as carriers of vaccines due to their ability to deliver the antigen and adjuvants to enhance the immune response. The site of vaccine administration plays a major role in the immune response to the vaccines [64]. The respiratory system is the entrance point of pathogens that may cause pulmonary diseases. Thus, the lungs retain all the necessary components to combat these diseases, granting protection against pulmonary transmitted antigens in addition to the large surface area of the lungs for interaction with antigens [65]. Those factors make the pulmonary system a good model for vaccine delivery. Pulmonary inhalation, intratracheal, and intranasal delivery systems may be used for vaccine delivery of local or systemic immunization based on the therapeutic intention [4].

##### Intranasal Delivery

The intranasal delivery route is effective, cost-effective, well-tolerated, and offers self-medication and convenient options to the patient. Drug delivery through the intranasal route is a novel and promising approach due to many factors, including the large epithelial surface area for drug absorption, avoiding first-pass metabolism through the direct passage of nanoparticles to the systematic circulation, rapid onset of action, and fewer side effects [66]. In addition, it demonstrates the capability of directly delivering nanoparticles to the brain via olfactory nerves [67]. The formulation of the drugs into nanoparticles facilitates their delivery through nasal cavity; for instance, colloidal formulation protects the nanoparticles from milieu degradation in the nasal cavity and enhances their absorption through mucosal barriers [68]. Vaccine delivery as nanoparticles intranasally has a beneficial effect on good immune responses, which is related to the fact that smaller particles absorbed more readily through lymphoid tissue in the nasal cavity. Nanoparticles should be formulated in a specific design that is suitable for intranasal vaccine delivery, taking into consideration the physiochemical properties of the drug, formulation, and the nasal physiological factors [69]. Nanocarriers have many unique characteristics that make them a good choice for vaccine delivery. Nanocarriers act as a vaccine adjuvant which increase the antigens that reach the immune system; thus, they combine the effects of antigen delivery systems and immuno-stimulation. Nanocarriers have the capacity of regulating the release of the antigen over extended intervals of time. Nanocarriers could be polymeric nanoparticles or lipid-based nanoparticles [70]. Chitosan nanoparticles are an example of polymeric nanocarrier that is used for influenza vaccine delivery intranasally. It was shown to be a good alternative to the parenteral route, which demonstrated an enhancement in mucosal immunity and ease in administration [71]. Moreover, chitosan is a mucoadhesive polymer that attaches to mucous membranes in the nasal cavity for a longer period and has less formulation clearance [72]. Despite the previously mentioned advantages of intranasal drug delivery, the intranasal route has some limitations, including the low dose and volume that can be administered, especially for low water-solubility drugs. This limitation can be avoided by using the right delivery device [66].

##### Intratracheal Delivery

Intratracheal drug administration is one of the old and commonly used routes, has the advantage of its low cost and its ability to deliver a well-defined dose to the lungs, as smaller amounts of drug are needed in comparison to aerosol administration method. The given dosage may be precisely determined and is not complicated by body site deposition or absorption. Despite the mentioned advantages, intratracheal instillation is relatively invasive and non-physiologic [73]. It is generally used in animal studies only rather than for clinical applications. The intratracheal route was used for in vivo delivery of peptides, nucleic acid, and drugs that are unstable in acidic highly enzymatic active media of the gastrointestinal system. Different nanoparticles can be used in an intratracheal delivery system including polymeric and lipid-based nanoparticles [74]. Intratracheal anthrax vaccine potency and efficacy were studied in vivo; the results showed higher lung mucosal and cellular immune response than that of the subcutaneous injection, which indicates that the intratracheal route maintains the stability and the effectiveness of the vaccine [75]. Moreover, when intratracheal immunization with inter-bilayer-crosslinked multilamellar vesicles (ICMVs), an antigen-carrying lipid nanocapsules vaccine, was evaluated in mice, the result showed induction in the transient inflammatory responses in the lungs with no tissue toxicity [76].

##### Nebulization Delivery

Aerosol vaccination is a non-invasive approach that mimics the physiological immunity induction after pathogens exposure. Nebulization provides an advantage of constant output of aerosol of large solutions and doses with little patient skills needed [77]. However, the generation of aerosol for the desired dose of nanomaterials is complex, expensive, and time-consuming. Pulmonary vaccination using nebulization is reported in several studies for live, attenuated pathogens such as measles, BCG, and rubella [78]. Synthetic nanoparticles, such as lipid nanoparticles, liposomes, polymeric nanoparticles, and hybrid nanoparticles, can be delivered via nebulizers for mucosal nano-vaccines. For example, aerosolized polymeric nanoparticles (PLGA and PLA) were used for pulmonary delivery of vaccine with greater immune response enhancement than free antigens [79]. However, nanoparticles interact with pulmonary cells and proteins differently from those in systemic delivery, and the shearing stress from nebulizers or inhalers may affect and disturb the structure of nanoparticles, especially the lipid-based nanoparticles. Thus, those factors should be taken into consideration while designing the nanoparticles formulation [80]. The measles vaccine is one of the most studied vaccines for pulmonary vaccination. Bennett JV et al. studied the immune response in 6-year-old children who receive a liquid aerosol vaccine, and the result showed a significant enhancement in the immune response compared with patients who received the vaccine through injection [81]. In contrast, the subcutaneous injection vaccine showed a significant enhancement in cellular and humoral immunity in a 9-month-old infant when compared with a patient who received the measles vaccine via liquid aerosol, which is related to difficulties in aerosol administration to a young infant [82]. Thus, in addition to formulation design problems, factors related to the efficiency of administration and its efficacy in humans should be considered.

#### 2.2.2. Polymeric Nanoparticles Delivery to the Lungs

Many different PNP delivery mechanisms have been developed using parenteral, oral, intraocular, transdermal, and pulmonary channels [30,83]. The pulmonary route can be used for either local or systemic drug delivery. Local administration of PNPs offers several advantages over conventional dosage forms such as high local drug concentration. This method also lowers systemic drug exposure, which reduces the number and risk of side effects, and prolongs drug residence time, especially in the case of bioadhesive PNPs. The advantages of PNP pulmonary drug delivery for systemic action are summarized in Figure 1. Non-invasive methods such as pulmonary drug delivery often show higher patient compliance compared with intravenous injection. Since PNPs can control and sustain the drug release, reduced dosing frequency is required [84]. The large alveolar surface area, the thin epithelial layer, the high vascularization area, the low proteolytic enzyme, and the hepatic metabolism avoidance offer good drug absorption, fast onset of action and high bioavailability [85,86,87]. Finally, aerosols (i.e., particulate drug carrier systems) are effective in delivering drugs for both local or systemic effects [88].

Many technologies are used for drug delivery to the lungs including pressurized metered-dose inhalers (MDIs), dry powder inhalers (DPIs) and jet or ultrasonic nebulizers [89]. Hand-bulb nebulizers initially gained popularity for supplying adrenaline chloride (bronchodilator). Afterwards, electric and ultrasonic nebulizers were developed [90]. In 1956, MDIs were introduced and rapidly overtook the market due to their smaller size, lower cost, easier use and faster, quieter delivery compared with nebulizer equipment. Currently, MDIs with high-dose drugs are used to develop inhalers and load local and systemic drugs including vaccines. Using MDIs requires the patient to create a sufficient flow rate to deliver the dose. Some patients are unable to accomplish this, and the desired therapeutic effect is not achieved [91]. To solve this problem, active DPIs were developed. Spiros^®^ platform (Dura Pharmaceuticals, Draper, UT, USA) for instance, uses a battery-operated propeller [92] while Exubera^®^ insulin inhalers (Pfizer, New York, NY, USA) utilize pressurized air for active powder dispersion from a hand piston [93]. These inhalers deliver fine particle doses with high flow rate to provide constant lung deposition and higher uniform distribution [89]. DPIs are also environmentally friendly, non-aqueous and user friendly. Since the drug is stored in a dry state, the drug’s stability is enhanced while maintaining sterility [94]. Unfortunately, active DPIs are complex, expensive and cumbersome in size.

Recently, a third generation of vaccines that are made of DNA and in vitro transcribed mRNA have been used instead of using inactivated or live, attenuated viruses. DNA and mRNA are simpler and can generate antigen-specific humoral and cellular immunity, without exposure to a real pathogen that invade the host [95]. Nucleic acid-based vaccines can be scaled up rapidly in infectious outbreaks [31]. Further, they have simple procedures of repeating culture and purification [96]. Lately, in coronavirus disease 2019 (COVID-19), mRNA vaccines enabled rapid vaccine development and protection against COVID-19 [97].

Delivery vehicles or devices used in the delivery of these nucleic acids are crucial for many reasons. First, DNA and mRNA have limited cellular uptake. Second, they are instable as they may degrade during administration [98]. Additionally, nucleic acid-based vaccines may cause cytokine storms, as well as cause many other side effects including allergy, renal failure, heart failure, and infarction [99].

Liposome nanoparticles were used to deliver both DNA and mRNA since they are stable particles. They usually consist of a cationic lipid bilayer shell, auxiliary lipids, cholesterol, and polyethylene glycol. The mRNA is encapsulated in the aqueous core of the liposomes to protect it from RNase [100]. Mostly, lipids are screened by pH-responsive materials because the cationic lipids are easily captured by immune cells [101]. Norbert et al. reported that mRNA encapsulated in lipid nanoparticles can induce high levels of TFH cells and GCB, generate effective neutralizing antibody response and antigen-specific CD4+ T cell response [102]. Polylysine was the first cationic polymer used successfully to deliver plasmid DNA in 1987 [101]. Since then, other polymers have been used for that purpose such as spermine, chitosan, polyethyleneimine, polyurethane, poly-amido-amine (PAA), polyethylenimine (PEI) and poly-beta amino-esters (PBAEs). For example, PEI was used to deliver DNA into the mouse brain to enhance siRNA or DNA transfection efficiency [103]. Further, it was employed as a carrier to introduce the HIV-1 gag gene into dendritic cells and BALB/c mice [104]. Other researchers prepared ferritin nanoparticle vaccine to deliver PreS1 to specific bone marrow cells. This vaccine was able to induce strong and persistent anti PreS1 effect in mice with hepatitis B [105].

Many factors impact the delivery of inhaled PNPs and their deposition in the lungs including particle size and distribution, particle morphology, electrical charge, hygroscopicity and particle density [106]. Herein, a description of the major factors affecting PNP formulation for pulmonary delivery is presented.

##### Particle Size

Inhaled NPs are deposited in the lung through impaction, sedimentation, interception and diffusion depending on the particle size [107]. These particles are usually characterized by their aerodynamic diameter (*d_a_*), which assumes that spherical particles settle under gravity. This diameter is calculated using Equation (1):
(1)$$d_{a}=\sqrt{(\frac{\rho }{\rho_{a}}}\times d_{g})$$
where ρ is the mass density of the particle, $\rho_{a}$ is the unit density (1 g/mL) and *d_g_* is the geometric diameter.

PNP deposition behavior can be predicted by the particles’ aerodynamic diameter. Deposition in the mouth and upper airways is expected for particles with *d_a_* greater than 5 μm. In the case of smaller particles with *d_a_* ranging between 1–5 μm, deposition in the lungs is likely. Very small particles with diameters less than 1 μm are expected to remain dispersed in air and then exhaled [108]. Therefore, the optimal size for a particle to deposit deep into the alveolar region has been defined as 1–3 μm. However, the bi-model deposition of particles in the lung allows for nanosized particles to settle effectively in the alveolar region with depositions up to 50% if slow inhalation and breath-holding are implemented [109]. The PNP size also affects post-deposition behavior. Optimally sized PNPs are critical to avoiding alveolar macrophage clearance and enhancing transepithelial transport. Small PNPs usually avoid macrophage clearance and show better transport [110]. The proteins and lipids that line the alveoli and act as a barrier against molecule absorption, the tight junction between the epithelial cell, the alveolar macrophages, and the transport mechanism of particles through active absorption or passive diffusion are all affected by PNP particle sizes [40,111].

##### Particle Morphology

The PNP particle shape affects clearance, adhesion, penetration, drug release, and targeting method. For instance, nanofibers exceeding 20 μm in length are too long to be phagocytosed and will be cleared very slowly from the body. Such nanofibers can remain in the lungs for months or, more likely, years [112].

The PNP shape impacts the hydrophilic or hydrophobic surface orientation, which controls their penetration and adhesion [113,114,115]. For instance, Lin et al. [116] reported that NPs of different shapes have different penetration capabilities, with rod-shaped NPs having the highest degree of penetration, and disruption ability of the surfactant monolayer covering respiratory tissues. The PNP transportation dynamics are related to the capillary force of the particles [117] as well as their orientation and rotation on the membrane surface [116], all of which is dictated by the shape of the nanoparticle.

##### Electrical Charge

Most PNPs carry electrical charges related to the ionization of the functional groups of the polymer used to prepare the nanoparticles or another coating material. The electrical charge of PNPs affects their endocytosis, adhesion, penetration, toxicity, drug delivery, and targeting. For example, the interaction between cells and lecithin-coated NPs involves cellular ligands that recognize the molecular charge of lecithin. It has also been reported that inhaled PNPs coated with lecithin or albumin have higher endocytosis in comparison with uncoated PNPs [118]. Additionally, positively charged NPs have higher adhesion efficiencies than charge-neutral NPs [31]. Polycationic PNPs showed strong interaction with cell membranes during in vitro studies [112]. Moreover, it was reported that the polar surfaces of NPs (high dielectric constants) have different translocation rates in hamsters during respiratory epithelium and circulation [119]. The toxicity of PNPs is also impacted by surface charge. For instance, Kemp et al. [120] showed that the administration of amine-modified polystyrene PNPs (positively charged with sizes of 400 nm) enhanced systemic thrombophilia for hamsters, while negatively charged PNPs of the same size did not.

## 3. Polymeric Nanoparticles Used for Inhaled Vaccination

### 3.1. Nanoparticles for Inhaled Vaccines

The extensive mucosal surface area presented in the lung makes the pulmonary system an effective system for vaccine delivery. This offers the advantage of lower dosage requirement and compatibility with a wide variety of antigens, including DNA- and RNA-based vaccines, in comparison with injection [121,122]. Most respiratory pathogens enter our body via mucosal membranes to cause an immune response, inducing excellent primary protection from pathogens [123,124]. The presence of related lymphatic tissues in the pulmonary system, including larynx, nasopharynx, and bronchi epithelium, induces a mucosal immune response and expands local defense mechanisms to the systemic defense mechanisms [125,126,127]. The physiology through the lung has favorable properties for vaccination that eliminates many issues faced by other mucosal systems including poor absorption, rapid clearance, degradation by antigens and enzyme tolerance [128].

As previously mentioned, DPIs are simple, cheap, compressed and disposable in a single unit, making them ideal for vaccine administration. Most antigens, including those used as vaccines, are macromolecules susceptible to chemical and/or physical degradation, especially in liquid formulations [129]. The delivery of such molecules as dry powder aerosols is a promising option expected to improve stability [130]. For example, dry powder measles vaccine and insulin formulations showed room temperature stability, eliminating the need for refrigeration [93,131,132]. In another example, live attenuated tuberculosis vaccine bacille Calmette–Gue’rin (BCG) was prepared by spray-drying. The BCG vaccine aerosol showed high efficiency in guinea pigs compared with animals immunized with parenteral BCG [133]. Another successful market formulation with a relatively low cost is the dry powder influenza vaccine, Inflexal V^®^ (Crucell) prepared using liposomal NPs [134]. PNPs have been used to load different antigens for vaccine applications via different mechanisms including covalent binding, adsorption and encapsulation [135].

In general, PNPs are used as an antigen carrier or as an adjuvant to stimulate immunity in both prophylactic and therapeutic applications [136,137,138]. Figure 2 shows different types of nanoparticles that are used for mucosal vaccine delivery. As a carrier, PNPs are loaded with the antigen, then target the immune cell. In this case, the antigen and/or carrier are engulfed by the immune cell or the PNPs release the antigen which is later engulfed by the immune cells [139]. For PNPs to behave as a carrier, assembly of the antigen and PNP is important. PNPs may induce certain immune phases, which then boost antigen identification and immunity stimulation [120]. Interactions between the antigen and PNPs are achieved by physical adsorption, chemical conjugation or encapsulation [139,140,141]. Antigen adsorption on the PNP surface is a simple but weak process dependent on charge or hydrophobic interaction. These interactions allow for rapid antigen separation from the NPs in vivo. In contrast, encapsulation and/or chemical conjugation form stronger bonds between the PNPs and antigen where the antigen is only released from the PNPs once the nanoparticles are destroyed [142]. In chemical conjugation, crosslinking the antigen and PNP surface is chemically achieved so that PNP is taken up by the cell. The antigen is then released inside the cell following chemical destruction of the covalent bond [143].

PNPs have been shown to work by themselves as immune stimulators or adjuvants. In this case, crosslinking between the PNP and antigen does not take place, and modification of the antigenic structure is possible when attached to the PNP interface [144]. Formulation of adjuvant PNPs with a specific antigen is possible by simply mixing the PNPs and the antigen directly before administration, which does not require strong association between the NPs and the antigen [145]. In general, when PNPs are the same size as the pathogen, they can be efficiently taken up by the antigen-presenting cells (APCs) to induce immune response [146]. It has been well established that dendritic cells generally uptake viruses with sizes ranging from 20 to 200 nm while macrophages uptake larger particles with sizes from 0.5 to 5 µm. In a study on polystyrene NPs, dendritic cells uptake particles with sizes of less than 500 nm [147]. In another study, PLGA NPs with a size of 300 nm were taken and activated by dendritic cells more favorably compared with particles larger than 1 μm [148].

Particle size is not the only factor that impacts PNP performance as an adjuvant in immunization. Other factors such as the preparation material are important. For example, amphiphilic poly (amino acid) (PAA) NPs 200 nm in size were taken up more effectively by dendritic cells than smaller ~30 nm NPs [149]. Surface charge also plays a major role in immune system stimulation [150,151]. PNPs with positive charge have been reported to cause higher uptake by APCs due to their stronger electrostatic interactions with the negatively charged cell membranes [152]. Positively charged polystyrene particles have been shown to be taken up by dendritic cells and macrophages more efficiently than neutral or negatively charged particles [147]. Furthermore, particle shape shows a dominant role in interactions between PNPs and APCs. It has been shown that the particle shape interface between particles and APCs affects macrophage phagocytosis [153]. For instance, Niikura et al. reported that spherical NPs are more effective in stimulating antibody response than cube and rod-shaped NPs, although the rod-shaped NPs were taken up more efficiently by APCs [154].

Next, this review will discuss the most popular polymer choices for PNP formulation, which include chitosan and polyesters (PLGA and PLA). Table 1 presents some of the polymeric nanoparticles that have been used in previous studies for intranasal vaccine delivery.

### 3.2. Chitosan and Chitosan Derivatives Nanoparticles

Chitosan is a polysaccharide composed of N-acetylglucosamine and glucosamine obtained via chitin n-deacetylation. Chitin is a biopolymer found in crustacean shells or fungi mycelium [163]. Chitosan is a non-toxic, biodegradable, and biocompatible cationic polymer. Because of these favorable biological characteristics, chitosan is used to deliver many pharmaceutical agents [164,165]. Chitosan also has interesting mucoadhesive properties that can stimulate immune system cells that has led to interest in its usage as an antigen vaccine carrier via the intranasal route [166]. The mucoadhesive properties of chitosan are primarily attributed to its amino groups that are easily protonated in weak acidic environments. This amine group interacts with the sialic acid moieties in mucin, the main protein component of mucus, which is negatively charged at physiological pHs by electrostatic forces [13].

Furthermore, chitosan nanoparticles (CS NPs) can protect vaccines from degradation via incorporation in the NP core. Chitosan has also been reported as an adjuvant for mucosal vaccination, especially when delivered intranasally. The mucoadhesiveness of chitosan allows the antigen to reside for longer durations at the mucosal surface, which is expected to enhance the APC antigen uptake. This effect has been shown to enhance immune cell stimulation [50].

It is important for the particles to penetrate mucus at a rate higher than the rate of mucosal renewal and clearance. Therefore, modifications of CS NPs were suggested to enhance its mucus-penetrating properties. One of these modifications is derivatization with poly (ethylene glycol) (PEG). PEG is a neutral hydrophilic polymer used to decorate CS NPs for many reasons including enhancing its mucus-penetrating capabilities in a process called PEGylation. PEGylation can minimize mucoadhesive interactions between CS and mucins, and thus allow rapid penetration through mucus [167]. The efficacy of PEG was related to its molecular weight as mentioned by Maisel et al.

Other researchers tried to enhance its mucus-penetrating properties by increasing the solubility of CS. For instance, Ways et al. synthesized four derivatives of CS (PEG, PHEA, POZ and PVP). The modified CS NPs showed enhanced and deeper muco-penetration into sheep nasal mucosa [168].

Additionally, both chitosan and CS NPs have been reported to produce a modulatory effect on the epithelial intercellular tight junction and increase paracellular drug transport [169]. Chitosan easily forms PNPs, making NP preparation a simple procedure where the NP size and charge can easily be tuned by controlling the experimental conditions and raw material properties [170]. CS NP formulation generally avoids excessive use of harmful organic solvent, which is often needed to enhance the entrapment or adsorption of therapeutic antigens and proteins [171]. Often, CS NPs are formed using cross-linking materials such as tripolyphosphate (TPP) to improve the encapsulation efficiency using the ionotropic gelation method [172].

The cationic nature of chitosan offers the advantage of carrying non-viral materials such as DNA for vaccination applications. Since nucleic acids have a strong negative charge, they can undergo electrostatic interaction with chitosan to form particulate entities known as polyplexes [173]. This interaction protects nucleic acids until they are delivered to the target site [169]. Bivas-Benita et al. showed that CS NPs are an ideal DNA vaccine delivery system due to their ability to protect the DNA from nucleases degradation and the enhanced immunity they provide by inducing dendritic cells maturation and increasing IFN-secretion from T cells after pulmonary immunization against tuberculosis [174].

CS NPs have also been used for mucosal vaccination of loaded antigen [13,175]. Live Newcastle virus loaded in CS NPs and delivered to chickens by intranasal and oral routes was shown to induce a higher IgA antibody response compared with chickens immunized with the plasmid control [176]. In other studies on recombinant pertussis toxin and *Bordetella pertussis* filamentous haemagglutinin loaded in CS NPs, very strong mucosal and systemic immune reactions were observed for nasal administration [177]. Further, systemic and local immune responses were induced after nasal administration of a diphtheria toxin mutant and CS in mice [178]. Other studies showed that influenza subunit virus vaccine delivered intranasally in CS NPs enhances both mucosal and systemic antibody and cell-mediated immune response in mice [159]. Furthermore, intranasal delivery of inactivated swine influenza A virus encapsulated in CS NPs enhanced mucosal antibody and cell-mediated immune responses in pigs. In this study, the intranasal vaccination improved the mucosal secretory IgA in the respiratory tract and regional lymph nodes. It also enhanced the systemic IgG and T-cell responses against different subtypes of swine influenza A virus [179]. Borges et al. reported that intranasal delivery of hepatitis B vaccine loaded in CS NPs enhances the mucosal IgA antibody response [180]. Other studies demonstrate that intranasal immunization using CS NPs as a vaccine carrier induces both mucosal and systemic antibody responses against Pneumococcus, Bordetella and Diphtheria species [177,181,182].

Both humoral and cell-mediated immune responses against Streptococcus zooepidemicus were achieved when CS NPs were delivered intranasally in mice [152]. Likewise, tetanus toxoid encapsulated in CS NPs delivered intranasally to rats was effectively transported across the nasal epithelium and produced mucosal and systemic antibody responses that last longer compared to solubilized antigen [183]. It has also been reported that mice immunized intranasally using CS NPs encapsulated with influenza split virus vaccine were able to induce a higher response than soluble antigens. In this study, both mucosal and systemic antibodies were enhanced as well as the cellular immune response indicated by increased IFNγ-secreting cell frequency in the spleen [159].

Some researchers are concerned with the effect of the physicochemical properties of chitosan used to prepare CS NPs loaded with vaccines on immune response. In the work of Vila et al., CS NPs loaded with tetanus toxoid formulated using high molecular weight chitosan (70 KDa) stimulated high IgG immune responses for extended periods after nasal administration to mice [184]. On the other hand, another study reported no detectable differences between IgA levels produced against tetanus toxoid entrapped in CS NPs synthesized using chitosan with different molecular weights, suggesting that the CS NP mode of action has not significantly impacted molecular weight [183].

However, chitosan faces major limitations for nano-drug carrier applications including poor solubility at physiological pH [185]. In response to these issues, several chitosan derivatives have been prepared to improve solubility and retain positive charge [186]. For example, N-trimethyl chitosan chloride (TMC), one of the most studied derivatives, shows good water solubility at physiological pH. TMC retains the mucoadhesive properties and excellent absorption-enhancing properties of chitosan [187,188]. Many studies showed successful use of TMC as an intranasal vaccine delivery system. In addition, NPs prepared using TMC with TPP as an ionic crosslinker increased nasal residence time of the encapsulated antigen, enhanced antigen uptake by M-cells and stimulated maturation of dendritic cells (DCs) [189,190].

Amidi et al. observed that intranasal administration of influenza antigen encapsulated in TMC-NPs significantly improved the systemic and mucosal immune response compared with intramuscular or intranasal administration of soluble influenza vaccine [191]. TMC particles loaded with antigens, e.g., tetanus toxoid (TT), have also been reported to induce mucosal and systemic antibody responses [192].

### 3.3. Polyesters: PLGA and PLA

PLGA is a copolymer of PGA and PLA. Both PLGA and PLA are FDA-approved polymers that are biodegradable, biocompatible and have been widely studied to design delivery systems for small drugs, peptides, proteins and other macromolecules such as RNA and DNA [193,194].

NPs prepared of PLGA, PLA and their derivatives are considered promising vaccine carriers. PLGA and PLA are believed to be able to improve antigen delivery to the immune system and to enhance mucosal surface interaction. PLGA and PLA increase epithelium penetration while maintaining full protection of the entrapped antigen and enhance antigen recognition by the mucosal immune system. PLGA and PLA can be used to achieve controlled release antigens in a predetermined manner [195,196]. All of these advantages make PLGA, PLA and their derivatives strong drug delivery candidates, especially for vaccine applications.

Hiremath et al. demonstrated that H1N1 influenza peptides encapsulated in PLGA NPs as a vaccine delivery system enhanced the virus-specific T cell response in pig lungs and reduced the virus load in their airways [197]. Furthermore, Mansoor et al. showed that bovine parainfluenza 3 virus antigen, encapsulated in PLGA NPs and administered intranasally to calves, had notably greater mucosal IgA responses compared with calves that received the commercially available respiratory vaccine [198]. The sustained immunological responses were attributed to the sustained antigen release from PLGA NPs in the nasal mucosa [198].

A combination of factors including hydrophobicity, particle size and polymer type play a vital role in generating mucosal immune response. Therefore, optimization of these variables is required. Many studies have been dedicated to the influence of particle size on immune response generated by the antigens encapsulated in polyester-based NPs following nasal administration. For example, the immune response to ovalbumin encapsulated by PLA NPs administered intranasally was found to be significantly greater than the response to PLA microparticles [199]. Moreover, it was reported that PLGA NPs with a ~200 nm size are optimal for dendritic cell interaction and induction of an effective cellular immune response [200]. Further, the ratio of PLA to PGA in PLGA was shown to affect the immune response and behavior of NPs when administered intranasally. In a study conducted by Thomas et al., intranasally delivered hepatitis B vaccine loaded in PLGA NPs with different ratios of PLA to PGA (PLGA 85:15 and PLGA 50:50). The results showed that the particle size and drug release increased with increasing glycolide monomer ratio, which led to decreased immune response [79].

The main obstacle for PLGA-based intranasal vaccine delivery is poor mucoadhesiveness. This limits the formulation residence time to less than 20 min. In most cases, this is not enough time for antigen uptake by APCs [201,202]. Many attempts to overcome this challenge focus on modifying or incorporating other polymeric particles with mucoadhesive properties on the surface of PLGA NPs to prolong residence time in the nasal cavity [201,203,204]. Highly mucoadhesive polymers such as chitosan or its derivatives, for instance, can be used to modify the properties of PLGA NPs. PEGylation of PLGA NPs was used to enhance the nanoparticles mucus-penetrating properties [205]. Nasal immunization with hepatitis B antigen-loaded in TMC-coated PLGA NPs significantly increased antigen-specific antibodies compared with nasally immunized mice with hepatitis B loaded in uncoated PLGA NPs [206]. In another study, only TMC-NPs loaded with ovalbumin increased the antigen nasal residence time compared with PLGA, TMC-NPs and TMC-coated PLGA NPs loaded with ovalbumin after intranasal delivery. This was attributed to the ability of TMC-NPs to stimulate dendritic cells maturation which led to enhanced antibody responses [207].

Pawar et al. designed a system of PLGA NPs encapsulated with hepatitis B antigen. Further, the NPs were coated with chitosan and glycol chitosan to improve the mucoadhesive properties. The results indicated glycol chitosan–PLGA NPs had a higher residence time in comparison with the chitosan-coated and uncoated PLGA NPs. This provided better systemic and local uptake of the antigen by immune cells and induced potent immune responses [201]. The improved mucoadhesive ability of glycol chitosan–PLGA NPs was related to the better mucin binding of glycol chitosan compared to chitosan [208].

## 4. Polymeric Nanoparticles as Vaccine Adjuvants

Usually, newly developed vaccines contained pure recombinant or synthetic antigens to reduce the adverse effect associated with live or killed organism vaccines. Unfortunately, these vaccines are less immunogenic and induce lower humoral and cellular responses [209]. To achieve a better immune response with the administration of synthetic antigens and prolong the protection period against infections, adjuvants are sometimes included within the formulation [210]. Adjuvants are essential immune-stimulating vaccine components that can either induce or boost immune response to antigens included in the vaccine formulation [211,212,213,214]. Moreover, they can act as a depot for the antigen to control or slow its release and decrease the amount of the antigen required to create robust immune response. Adjuvants can modify the immune response by targeting APCs and provide risk signals that help the immune system respond to the antigen [210,215,216]. There are two categories of adjuvants used in vaccination. The first are immunostimulatory compounds such as cytokines, bacterial toxins, and toll-like receptors (TLRs). These molecules stimulate immune responses as they interact with specific receptors. The second are delivery systems that can protect entrapped antigens within them from degradation and/or enhance immune response due to their specific characteristics. Examples of these include nanoparticles, liposomes and other particles [217]. In general, an ideal adjuvant should have good biocompatibility, biodegradability, and stability, as well have low prices and a long shelf life [210].

Some polymers used to prepare PNPs are considered vaccine adjuvants. Therefore, some polymers have attracted more interest in the formulation of mucosal vaccines because they can significantly enhance humoral, cellular, and mucosal immune response. Currently, both natural and synthetic polymeric NPs are employed in vaccine formulation studies as adjuvants [8,218].

Chitosan is one such polymer that has mucoadhesive properties and exceptional ability to open tight junctions between epithelial cells [219]. Chitosan was shown to promote antigen transport through various antigen-delivery pathways, stimulating a powerful mucosal immune response [220]. McNeela et al. reported that chitosan increases immunogenicity of diphtheria toxoid (DT) vaccine administered nasally where it can induce high levels of secretory IgA, Ag-specific IgG, toxin-neutralizing antibodies and T cell responses, particularly T-helper cells 2 subtype [181]. Additionally, the bioadhesive character and intrinsic adjuvanticity of CS NPs related to the chitosan-mediated inflammasome activation made it suitable as a vaccine adjuvant delivery system [221]. Finally, it is worth mentioning that chitosan derivatives are also considered adjuvants that have shown even better responses compared with native chitosan [222].

PLGA NPs enhance antigen recognition by APCs and enable antigen processing and presentation to naïve lymphocytes [223,224]. PLGA-based NPs liberate encapsulated antigens to APCs in a controlled and sustained manner over a long period of time. This has been shown to be an important role for PLGA NPs as vaccine adjuvants where they induce mucosal and systemic immune responses to antigens [225].

## 5. Challenges of Polymeric Nanoparticles Delivery for Inhaled Vaccination

PNPs used to deliver pharmaceutical agents to the lungs, including vaccines, are facing several challenges. Formulation instability and reproducibility are among the most important challenges. The physicochemical properties affect antigen-loaded PNP interactions with immune cells, influencing the immunological outcome. The shape, size, polydispersity, surface charge, hydrophobicity, stability and bioadhesive properties of the PNPs require stabilization and reproducibility [226]. Additional factors related to polymers used in PNP preparation include crystallinity, glass transition temperature and bioadhesiveness [227,228]. Antigen release typically correlates with amorphous polymer phases. Moreover, it was noted that crystalline PNPs loaded with antigen may lower the release rate [193]. Additionally, attention must be dedicated to the adverse effects of PNPs on immune cells, such as immune-mediated destruction or rejection that led to PNP elimination and immune toxicity [44,229].

There are other limitations related to PNP delivery devices loaded with vaccines. For instance, powder vaccines may cause high capsule volume causing device retention, which in turn leads to failed vaccine delivery. Other safety issues must be accounted for, especially in high-risk children with asthma and immunodeficient patients [44]. Other factors are related to the importance of inhalation control to ensure effective drug delivery to the lung, such as holding the breath, patient age (elder patients usually experience larger residual lung volumes than the alveolar volume) and breathing frequency [230].

In general, the industrial scale-up of pulmonary vaccine faces restricted instructions and regulatory parameters. These regulations become tighter when NPs are involved in the formulation. One of the major concerns is the thermal sensitivity of PNPs loaded with vaccines that must be thoroughly studied. Further, temperature control may increase the product price. Therefore, thermally stable PNPs are required to overcome the thermal instability problem, and hence decrease the cost of the final product [225]. Many researchers are trying to formulate these dosage forms as powders. Powders are generally cheaper, more stable and more immunogenic in comparison with liquid nebulizers [231]. Finally, the overall price of the formulation and the devices used to deliver them are higher than other drug delivery systems [232].

## 6. Conclusions

Inhaled vaccines have recently earned significant attention as an effective and non-invasive alternative for intravenous vaccination. The inhaled vaccine’s ability to induce both mucosal and systemic immune responses is a key feature, as most pathogens enter the body through the mucosal systems. PNPs such as CS NPs, PLGA NPs and PLA NPs offer the advantage of biocompatibility and biodegradability as well as antigen protection. PNPs have also successfully showed adjuvant properties that further enhance the immune response. Unfortunately, many problems must be solved before PNP pulmonary vaccination becomes available to the market. As such, loading vaccines in PNPs can solve many health care-related problems and advance the field, but not before overcoming many limitations.

## Figures and Tables

**Figure 1 polymers-14-04450-f001:**
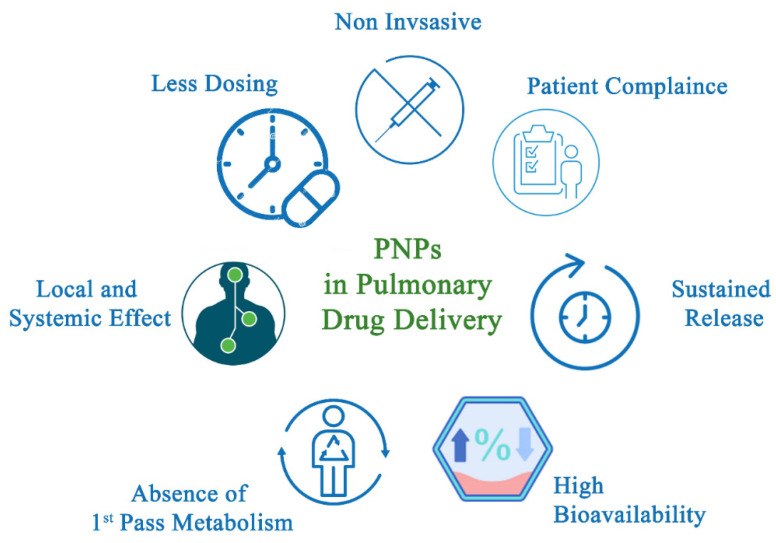
Advantages of PNPs in Pulmonary Drug Delivery Systems.

**Figure 2 polymers-14-04450-f002:**
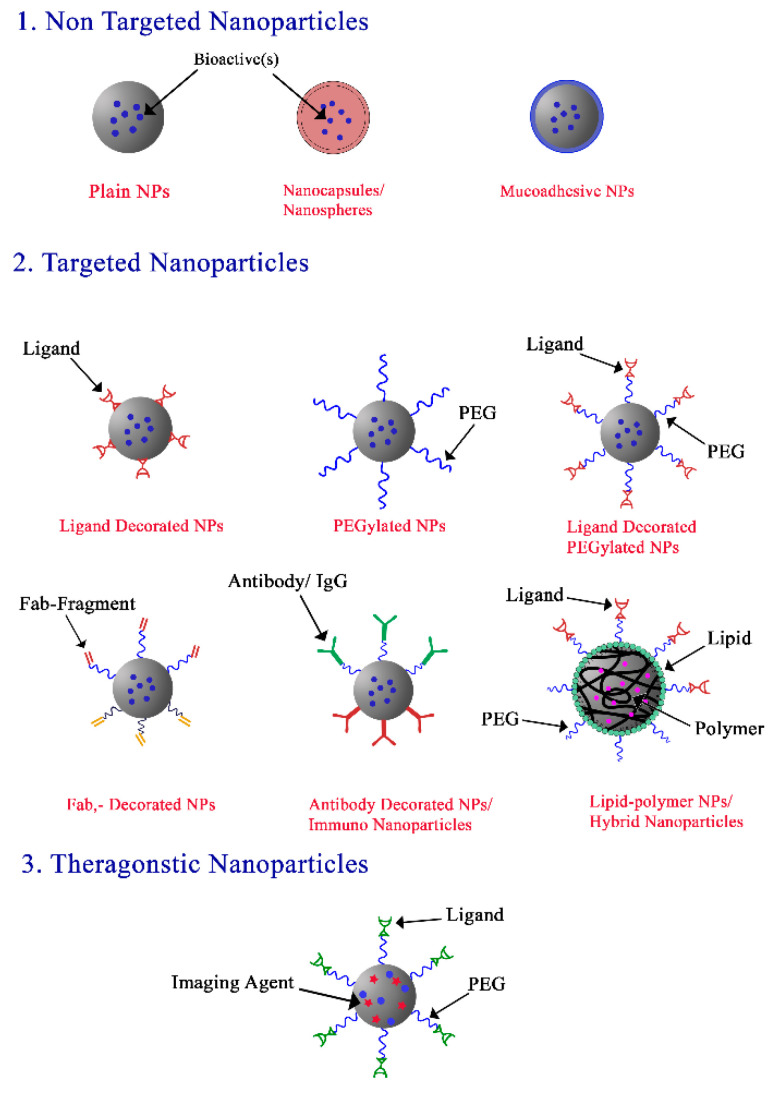
Different types of nanoparticles that are used for mucosal vaccine delivery.

**Table 1 polymers-14-04450-t001:** Examples of polymeric nanoparticles used for intranasal vaccine delivery.

The Composition of Nanoparticles	Antigen	Type of Immunity	Reference
PLGA ^(1)^ (50:50)	Synthetic bovine parainfluenza virus type-3 (BPI3V) peptide motifs	Induce stronger IgG ^(2)^ antibody	[155]
PLGA	Inactivated PRRS ^(3)^ virus	IgG1 and IgG2 antibody, T-helper (Th)-1 and Th2 ^(4)^ cytokines	[156]
Chitosan (mannose)	Tumor pGRP ^(5)^ DNA	Anti-GRP IgG antibody	[157]
Mannosylated chitosan	Foot and mouth disease virus DNA, pVAC FMDV ^(6)^ VP1-OmpA	Induction of virus-neutralizing antibodies, Th1(IgG2) and Th2 (IgG1) responses	[158]
CS/TPP ^(7)^	Subunit/split influenza	Higher systemic and mucosal antibody	[159]
PLGA coated gelatin (PGNPs) ^(8)^	Tetanus Toxoid antigen	Humoral, cellular and mucosal immunity	[160]
Chitosan dextran sulfate	Pertussis toxin (PTX)	-	[161]
Trimethylated chitosan (TMC) and chitosan	Hepatitis B surface antigen	IgG, IgG1, IgG2a, IgA ^(9)^ antibodies	[162]

^(1)^ Poly lactic-co-glycolic acid, ^(2)^ immunoglobulin G, ^(3)^ porcine reproductive and respiratory syndrome, ^(4)^ T helper type 1&2, ^(5)^ peptidoglycan recognition protein, ^(6)^ foot-and-mouth disease virus, ^(7)^ chitosan/tripolyphosphate, ^(8)^ polymer-grafted nanoparticles, ^(9)^ immunoglobulin A.

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
