# Peer review of "Polymeric Nanoparticles for Inhaled Vaccines"

_polymers, 2022, doi:10.3390/polym14204450_

Round 1

Reviewer 1 Report

Comments on polymers-1933535

In this manuscript, the authors reviewed polymeric nanoparticles that have been developed for vaccine delivery through the pulmonary routes. However, this manuscript needs to address several issues.

1.       The contents of this review are not up-to-date. It is noticed that only 8 papers published in recent 3 years were cited. The authors should do an extensive search and review recent papers.

2.       This manuscript did not include polymeric nanoparticles that were developed for nucleic acid (e.g., DNA and RNA) delivery. Nucleic acid is a more common payload used for vaccine delivery and thus it should not be ignored.

3.       This manuscript did not describe the mechanism of nanoparticle delivery through the pulmonary routes. How the nanoparticle interacts with the mucus, epithelial cells and antigen-presenting cells in the lung should be reviewed in detail.

4.       This manuscript did not discuss the administration methods for lung delivery. The difference between intranasal, intratracheal, and inhalation (nebulization) should be discussed. The nanoparticle design criteria for each administration route should be discussed.

5.       This manuscript focused on mucus-adhesive nanoparticles and failed to review mucus-penetrating nanoparticles. The mucus-penetrating nanoparticles are more recently developed, and they offer very different properties compared to the traditional mucus-adhesive ones. This needs to be included as well.

Author Response

Reviewer 1:

In this manuscript, the authors reviewed polymeric nanoparticles that have been developed for vaccine delivery through the pulmonary routes. However, this manuscript needs to address several issues.

Response:

Dear Reviewer (1),

Thank you for your comments that significantly improved the quality of the manuscript. Please find the replies to your comments. Hopefully, it is ready to be considered to publish.

  1. The contents of this review are not up-to-date. It is noticed that only 8 papers published in recent 3 years were cited. The authors should do an extensive search and review recent papers.

Response: References were updated, where around 100 references from the last 10 years were added.

  1. This manuscript did not include polymeric nanoparticles that were developed for nucleic acid (e.g., DNA and RNA) delivery. Nucleic acid is a more common payload used for vaccine delivery and thus it should not be ignored.

Response: Thank you, the role of polymeric nanoparticles in delivering vaccines made of nucleic acids was explained and the following was added to page 8:” Recently, a third generation of vaccines that are made of DNA and in vitro transcribed mRNA have been used instead of using inactivated or live, attenuated viruses. DNA and mRNA are simpler and can generate antigen-specific humoral and cellular immunity, without exposure to a real pathogen that invade the host.[95] Nucleic acid based vaccines can be scaled up rapidly in infectious outbreaks.[96] Further, they have simple procedures of repeating culture and purification.[97] Lately, in coronavirus disease 2019 (COVID-19), mRNA vaccines enabled rapid vaccine development and protection against COVID-19.[98]

Delivery vehicles or devices used in the delivery of these nucleic acids are crucial for many reasons. First, DNA and mRNA have limited cellular uptake. Second, they are instable as they may degrade during administration.[99] Additionally, nucleic acids based vaccines may cause cytokine storms, as well as having many other side effects including allergy, renal failure, heart failure, and infarction.[100]

Liposome nanoparticles were used to deliver both DNA and mRNA since they are stable particles. They are usually made of a cationic lipid bilayer shell, auxiliary lipids, cholesterol, and polyethylene glycol. The mRNA is encapsulated in the aqueous core of the liposomes to protect it from RNase.[101] Mostly, lipids are screened by pH-responsive materials because the cationic lipids are easily captured by immune cells.[102] Norbert et al reported that mRNA encapsulated in lipid nanoparticles can induce high levels of TFH cells and GCB, generate effective neutralizing antibody response, and antigen-specific CD4+ T cell response.[103] Polylysine was the first cationic polymer used successfully to deliver plasmid DNA in 1987.[102] Since then, other polymers have been used for that purpose such as spermine, chitosan, polyethyleneimine, polyurethane, poly-amido-amine (PAA), polyethylenimine (PEI) and poly-beta amino-esters (PBAEs). For example, PEI was used to deliver DNA into the mouse brain to enhance siRNA or DNA transfection efficiency.[104] Further, it was employed as a carrier to introduce the HIV-1 gag gene into dendritic cells and BALB/c mice.[105] Other researchers prepared ferritin nanoparticle vaccine to deliver PreS1 to specific bone marrow cells. This vaccine was able to induce strong and persistent anti PreS1 effect in mice with hepatitis B.[106]

  1. This manuscript did not describe the mechanism of nanoparticle delivery through the pulmonary routes. How the nanoparticle interacts with the mucus, epithelial cells and antigen-presenting cells in the lung should be reviewed in detail.

Response: Section 2.2 Nanoparticles drug delivery to the lungs was added to cover this point starting from page3 to page 5 as follow:

2.2. Nanoparticles drug delivery to the lungs

The delivery of particles to the different regions of the lungs depends on the particle size of the formulation. Based on the particle size, there are three different mechanisms of drug deposition through the pulmonary route, impaction, sedimentation, and diffusion.[40]

In impaction, the aerosol particles go through the oropharynx and upper respiratory passages at a very high velocity. The particles then interact with the respiratory wall and are deposited in the oropharynx regions.[41] This mechanism can be observed with particle sizes greater than 5 µm mainly in dry powder inhalation (DPI) and metered dose inhalators (MDI).[42] In the DPI, the deposition is mainly affected by the inspiratory effort of the patient. If the force of inhalation is insufficient, the dry powder will be deposited in the upper airways due to the mass of the particles.[42] In the MDI high particle sizes also tend to lead to the deposition of the particles in the upper respiratory tract region despite the high speed of the generated aerosol.[42,43]

Gravitational forces are mainly responsible for the second mechanism, which is particle sedimentation. Particles with certain mass and sizes between 1 to 5 µm are deposited in the smaller airways and bronchioles.[44] Sedimentation is also influenced by the breathing mechanism, Slow breathing patterns provide a sufficient period for efficient sedimentation.[45]

Also, the diffusion process plays a major role in the deeper alveolar areas of the lungs. The Brownian motion of the surrounding molecules in the aqueous lung surfactant causes a random movement of the particles that leads to the dissolution of the drug in alveolar fluid when contact with the lung surfactant which is essential for diffusion.[46] In addition, the diffusion process is also affected by concentration gradient.[47] Particles smaller than one to 0.5 µm are mainly deposited in the alveolar region, while most of the particles, because of their smaller sizes, are exhaled.[47]

Moreover, depending on the location of deposition which is mainly affected by the particle size,[43] the nanoparticles can interact with different cell types within the respiratory tract, such as epithelial cells and antigen-presenting cells.

The epithelial cells are tightly connected by intercellular junctions called tight junctions, nanoparticles can pass the respiratory epithelia by two different pathways, through tight junctions between the cells or transcellular by endocytosis.[48]

The firmly sealed tight junctions in the epithelial cells make a barrier for particle permeation.[49] Moreover, the mucus layer, which covers the upper and central respiratory tract, as well as the clearance process in these regions create more barriers that reduce the uptake of nanoparticles in the respiratory lumen.[49] Therefore, agents, called penetration enhancers, which reversibly open the tight junctions are added to the formulations to enhance the transport of particles to the systemic and/ or lymphatic circulation.[50] On the contrary, in the distal airway’s epithelium, just before the alveoli, the tight junctions between the epithelial cells are loose and particles up to 22 kDa can passively diffuse via paracellular pathways.[50]

In addition to the paracellular route by which relatively small proteins are absorbed, larger proteins can be taken up from the respiratory tract by the transcellular pathway, which includes both nonspecific and specific (receptor-mediated) endocytosis.[51] The transport of antibodies and plasma proteins such as albumin, across the epithelial cells occurs by receptor-mediated endocytosis.[52] On the other hand, it has been shown that macromolecular therapeutics pass the epithelial cells by nonspecific endocytosis.[51]

Moreover, there are a variety of immune cell populations in the lungs, such as phagocytic cells (macrophages) and antigen-presenting cells (dendritic cells DCs).[53] The main role of immature DCs, primarily within the mucosal tissue, is to recognize antigens by their protrusions into the airway or alveolar lumen.[54] In this mechanism, depending on the nature of the antigen, DCs get activated by their recognition receptors and enter the maturation process, once maturated, DCs rapidly migrate to the lymph nodes.[55] In the case of maturation, antigens are processed by the DCs and presented to naive T cells by the major histocompatibility complex (MHC II) in combination with upregulated co-stimulatory molecules on the DCs surface (CD40, CD80, CD86), and the release of cytokines.[55,56] The type and combination of the cytokines released will determine the nature of effector T cells induced (Th1, Th2, Treg, and Th17).[55,57]

NPs based vaccination protocols, that mainly target DCs, are efficient and promising strategies for the induction and enhancement of immune responses for cancer treatment,[58,59] in addition to viral and microbial infections.[60,61] NPs can extend the antigen exposure to immune cells, facilitate antigen capture by DCs and initiate antigen presentation pathways within these cells, thus enhancing T cell-mediated immune responses.[62] Also, NPs delivery system could be used to enhance DCs activation by triggering cell-surface molecules, this mechanism enhances the internalization of NPs by DC and thus induce immune activation with separate and specific DCs-activating signaling pathways. [63]

  1. This manuscript did not discuss the administration methods for lung delivery. The difference between intranasal, intratracheal, and inhalation (nebulization) should be discussed. The nanoparticle design criteria for each administration route should be discussed.

Response: The sections 2.2.1, 2.2.2, 2.2.3 were added starting from page 5 to page 7 as follow:”

2.2.1. Nanoparticles through different pulmonary routes of administration

Nanoparticles are of great interest as carriers of vaccines due to their ability to deliver the antigen and adjuvants, to enhance the immune response. The site of vaccine administration plays a major role in the immune response to the vaccines.[64] The respiratory system is the entrance point of pathogens that may cause pulmonary diseases. Thus, the lungs have all the necessary components to combat these diseases, granting protection against pulmonary transmitted antigens, in addition to the large surface area of the lungs for interaction with antigens.[65] Those factors make the pulmonary system a good model for vaccine delivery. Pulmonary inhalation, intratracheal, and intranasal delivery systems may be used for vaccine delivery of local or systemic immunization based on the therapeutic intention.[4]

  • Intranasal delivery

The intranasal delivery route is effective, cost-effective, well tolerated, and offers self-medication and convenient options to the patient. Drug delivery through the intranasal route is a novel and promising approach due to many factors, including the large epithelial surface area for drug absorption, avoiding first-pass metabolism through the direct passage of nanoparticles to the systematic circulation, rapid onset of action, and fewer side effect.[66] In addition, the capability of directly delivering nanoparticles to the brain via olfactory nerves.[67] The formulation of the drugs into nanoparticles facilitates their delivery through nasal cavity, for instance; colloidal formulation protects the nanoparticles from milieu degradation in the nasal cavity and enhances their absorption through mucosal barriers.[68] Vaccine delivery as nanoparticles intranasally has a beneficial effect on good immune responses, this is related to smaller particles absorbed more readily through lymphoid tissue in the nasal cavity. Nanoparticles should be formulated in a specific design that is suitable for intranasal vaccine delivery, taking into consideration the physiochemical properties of the drug, formulation, and the nasal physiological factors.[69] Nanocarriers have many unique characteristics that make them a good choice for vaccine delivery. Nanocarriers act as a vaccine adjuvant which increase the antigens that reach the immune system; thus, they combine the effects of antigen delivery systems and immuno-stimulation. Nanocarriers have the capacity of regulating the release of the antigen over extended intervals of time. Nanocarriers could be polymeric nanoparticles or lipid-based nanoparticles.[70] Chitosan nanoparticles are an example of polymeric nanocarrier that is used for influenza vaccine delivery intranasally, it was shown to be a good alternative to the parenteral route, which demonstrated an enhancement in mucosal immunity and ease in administration. [71] Moreover, chitosan is a mucoadhesive polymer that attaches to mucous membranes in the nasal cavity for a longer period and has less formulation clearance. [72] Although the previously mentioned advantages of intranasal drug delivery, the intranasal route has some limitations, including the low dose and volume that can be administered, especially for low water solubility drugs. This limitation can be avoided by using the right delivery device. [66]

  • Intratracheal delivery

Intratracheal drug administration is one of the old and commonly used routes, has the advantage of its low cost and its ability to deliver a well-defined dose to the lungs, smaller amounts of drug needed relative to aerosol. The given dosage may be precisely determined and is not complicated by body site deposition or absorption. Although the mentioned advantages, intratracheal instillation is relatively invasive and non-physiologic.[73] It is generally used in animal studies only rather than for clinical applications. The intratracheal route was used in delivering peptides, nucleic acid, and drugs in vivo, that are unstable in acidic highly enzymatic active media of the gastrointestinal system. Different nanoparticles system can be used in an intratracheal delivery system including polymeric and lipid-based nanoparticles.[74] Intratracheal anthrax vaccine potency and efficacy were studied in vivo, the results showed higher lung mucosal and cellular immune response than the subcutaneous injection, this indicates that the intratracheal route maintains the stability and the effectiveness of the vaccine.[75] Moreover, intratracheal immunization with inter-bilayer-crosslinked multilamellar vesicles (ICMVs), an antigen-carrying lipid nanocapsules vaccine was evaluated in mice, the result showed induction in the transient inflammatory responses in the lungs with no tissue toxicity.[76]

  • Nebulization delivery

Aerosol vaccination is a non-invasive approach, that mimics the physiological immunity induction after pathogens exposure. Nebulization provides an advantage of constant output of aerosol of large solutions and doses with little patient skills needed.[77] However, the generation of aerosol for the desired dose of nanomaterials is complex, expensive, and time-consuming. Pulmonary vaccination using nebulization is reported in several studies for live, attenuated pathogens such as measles, BCG, and rubella.[78] Synthetic nanoparticles varying from lipid nanoparticles, liposomes, polymeric nanoparticles, and hybrid nanoparticles, can be delivered via nebulizers for mucosal nano-vaccines. For example, aerosolized polymeric nanoparticles (PLGA and PLA) were used for pulmonary delivery of vaccine, with greater immune response enhancement than free antigens. [79] However, nanoparticles interact with pulmonary cells and proteins differently from those interact in systemic delivery, and the shearing stress from nebulizers or inhalers may affect and disturb the structure of nanoparticles, especially the lipid-based nanoparticles, thus, those factors should be taken into consideration while designing the nanoparticles formulation. [80] The measles vaccine is one of the most studied vaccines for pulmonary vaccination. Bennett JV et al. study the immune response in 6-year-old children who receive a liquid aerosol vaccine, and the result showed a significant enhancement in the immune response compared to patient who receive the vaccine through injection.[81] In contrast, the subcutaneous injection vaccine showed a significant enhancement in cellular and humoral immunity in a 9-month-old infant, when compared to a patient who receive the measles vaccine via liquid aerosol, which is related to difficulties in aerosol administration in a young infant.[82] Thus, in addition to formulation design problems, factors related to the efficiency of administration and its efficacy in humans should be considered.

  1. This manuscript focused on mucus-adhesive nanoparticles and failed to review mucus-penetrating nanoparticles. The mucus-penetrating nanoparticles are more recently developed, and they offer very different properties compared to the traditional mucus-adhesive ones. This needs to be included as well.

Response: The mechanisms used to enhance the mucus-penetrating properties of chitosan and PLGA were added to the pages 13 & 15, respectively as follow:

“It is important for the particles to penetrate mucus at a rate higher than the rate of mucosal renewal and clearance. Therefore, modifications of CS NPs were suggested to enhance its mucus-penetrating properties. One of these modification is derivatization with Poly(ethylene glycol) (PEG). PEG is a neutral hydrophilic polymer, used to decorate CS NPs for many reasons including enhancing its mucus-penetrating capabilities in a process called PEGylation. PEGylation can minimize mucoadhesive interactions between CS and mucins, and thus allow rapid penetration through mucus.[169] The efficacy of PEG was related to its molecular weight as mentioned by Maisel et al.

Other researchers tried to enhance its mucus-penetrating properties by increasing the solubility of CS. For instance, Ways et al synthesized four derivatives of CS (PEG, PHEA, POZ and PVP). The modified CS NPs showed enhanced and deeper muco-penetration into sheep nasal mucosa.[170].”

“PEGylation of PLGA NPs was used to enhance the nanoparticles mucus-penetrating properties”

Reviewer 2 Report

The manuscript is well organize and original. The Abstract , Introduction and Experimental details are good. All the Figures are clear and the references are related to the work. So. i recommended that the paper is accepted to be published in polymers journal.

Author Response

Reviewer 2

Dear Reviewer (2),

Thank you.

The manuscript is well organize and original. The Abstract , Introduction and Experimental details are good. All the Figures are clear and the references are related to the work. So. i recommended that the paper is accepted to be published in polymers journal.

Reviewer 3 Report

1-The abstract is a comprehensive summary of the whole article. The abstract section is more introductive and needs to be revised by incorporating some scientific innovation and methodology.

2- Replace "inhaled" in the keywords with "inhaled vaccine".

3-The introduction section is well-written; however, needs to discuss more of the research published articles. such as Journal of Controlled Release (2022), 341,132-146. and International Journal of Biological Macromolecules (2022), 203, 222-243. 

4- Some references are too old (before 2010), replace them with some newly published papers.

5- The authors should prepare Figure 2 with better resolution.

Author Response

Reviewer 3

Dear Reviewer (3),

Thank you for your comments that significantly improved the quality of the manuscript. Please find the replies to your comments. Hopefully, it is ready to be considered to publish.

1-The abstract is a comprehensive summary of the whole article. The abstract section is more introductive and needs to be revised by incorporating some scientific innovation and methodology.

Response: The abstract was updated as follow: “Abstract: Many recent studies focus on the pulmonary delivery of vaccines as it is needle-free, safe, and effective. Inhaled vaccines enhance systemic and mucosal immunization but still faces many limitations that can be resolved using polymeric nanoparticles (PNPs). This review focuses on the use of properties of PNPs, specifically chitosan and PLGA to be used in the delivery of vaccines by inhalation. It also aims to highlight that PNPs have adjuvant properties by themselves that induce cellular and humeral immunogenicity. Further, different factors influence the behavior of PNP in vivo such as size, morphology, and charge are discussed. Finally, some of the primary challenges facing PNPs were discussed including formulation instability, reproducibility, device-related factors, patient-related factors, and industrial-level scale-up. Herein, the most important variables of PNPs that shall be defined in any PNPs to be used for pulmonary delivery are defined. Further, it focuses on the most popular polymers used for this purpose.”

2- Replace "inhaled" in the keywords with "inhaled vaccine".

Response: Done.

3-The introduction section is well-written; however, needs to discuss more of the research published articles. such as Journal of Controlled Release (2022), 341,132-146. and International Journal of Biological Macromolecules (2022), 203, 222-243. 

Response: These two references were added and many others through the paper.

4- Some references are too old (before 2010), replace them with some newly published papers.

Response: References were updated, where around 100 reference from the last 10 years were added.

5- The authors should prepare Figure 2 with better resolution.

Response: Done

Reviewer 4 Report

The paper can be published without change it

Author Response

Reviewer (4),

The paper can be published without change it

Response:

Dear Reviewer (4),

Thank you.

The paper can be published without change it

Thank you for your consideration.

Sincerely,

Nusaiba Al-Nemrawi, PhD

Associate Professor, Department of pharmaceutical Technology

Jordan University of Science and Technology

Round 2

Reviewer 1 Report

In the revised manuscript, all my previous comments have been addressed. Therefore, I recommend acceptance of this manuscript.